# Disentangling Style and Content in Anime Illustrations

## Abstract

Existing methods for AI-generated artworks still struggle with generating high-quality stylized content, where high-level semantics are preserved, or separating fine-grained styles from various artists. We propose a novel Generative Adversarial Disentanglement Network which can disentangle two complementary factors of variations when only one of them is labelled in general, and fully decompose complex anime illustrations into style and content in particular. Training such model is challenging, since given a style, various content data may exist but not the other way round. Our approach is divided into two stages, one that encodes an input image into a style independent content, and one based on a dual-conditional generator. We demonstrate the ability to generate high-fidelity anime portraits with a fixed content and a large variety of styles from over a thousand artists, and vice versa, using a single end-to-end network and with applications in style transfer. We show this unique capability as well as superior output to the current state-of-the-art.

## 1 Introduction

Computer generated art (Hertzmann, 2018) has become a topic of focus lately, due to revolutionary advancements in deep learning. Neural style transfer (Gatys et al., 2016) is a groundbreaking approach where high-level styles from artwork can be re-targeted to photographs using deep neural networks. While there has been numerous works and extensions on this topic, there are deficiencies in existing methods. For complex artworks, the methods that rely on matching neural network features and feature statistics, do not sufficiently capture the concept of style at the semantic level. Methods based on image-to-image translation (Isola et al., 2017) are able to learn domain specific definitions of style, but do not scale well to a large number of styles.

In addressing these challenges, we found that style transfer can be formulated as a particular instance of a general problem, where the dataset has two complementary factors of variation, with one of the factors labelled, and the goal is to train a generative network where the two factors can be fully disentangled and controlled independently. For the style transfer problem, we have labelled style and unlabelled content.

Based on various adversarial training techniques, we propose a solution to the problem and call our method Generative Adversarial Disentangling Network. Our approach consists of two main stages. First, we train a style-independent content encoder, then we introduce a dual-conditional generator based on auxiliary classifier GANs.

We demonstrate the disentanglement performance of our approach on a large dataset of anime portraits with over a thousand artist-specific styles, where our decomposition approach outperforms existing methods in terms of level of details and visual quality. Our method can faithfully generate portraits with proper style-specific shapes and appearances of facial features, including eyes, mouth, chin, hair, blushes, highlights, contours, as well as overall color saturation and contrast. To show the generality of our method, we also include results on the NIST handwritten digit dataset where we can disentangle between writer identity and digit class when only the writer is labelled, or alternatively when only the digit is labelled.

## 2 BACKGROUND

**Neural Style Transfer.** Gatys et al. Gatys et al. (2016) proposed a powerful idea of decomposing an image into content and style using a deep neural network. By matching the features extracted from one input image using a pre-trained network and the Gram matrix of features of another image, one can optimize for an output image that combines the content of the first input and the style of the second one. One area of improvement has been to first identify the best location for style source in the style image for every location in the content image. Luan et al. Luan et al. (2017) use masks from either user input or semantic segmentation for guidance. Liao et al. Liao et al. (2017) extends this approach by finding dense correspondences between the images using deep neural network features in a coarse-to-fine fashion. In another line of research, different representations for style have been proposed. In Huang et al. Huang and Belongie (2017), the style is represented by affine transformation parameters of instance normalization layers. These methods go slightly beyond transferring texture statistics. While we agree that texture features are an important part of style, these texture statistics do not capture high-level style semantics. Our observation is that, the concept of "style" is inherently domain-dependent, and no set of features defined a priori can adequately handle style transfer problem in all domains at once, let alone mere texture statistics. In our views, a better way of addressing style transfer would be to pose it as an image-to-image translation problem.

**Image-to-image Translation.** Isola et al. introduced the seminal work on image-to-image translation in (Isola et al., 2017) where extensive source and target training data need to be supplied. Several extensions to this approach, such as CycleGAN (Zhu et al., 2017) and DualGAN (Yi et al., 2017), removed the need for supervised training, which significantly increases the applicability of such methods. Of particular interest, Zhu et al. Zhu et al. (2017) demonstrates the ability to translate between photorealistic images and the styles of Van Gogh, Monet and Ukiyo-e. While the results are impressive, a different network is still required for each pair of domains of interest. One solution to this, is to train an encoder and a generator for each domain such that each domain can be encoded to and generated from a shared code space, as described in Liu et al. Liu et al. (2017). We wish to take one step further and use only one set of networks for many domains: the same encoder encodes each domain into a common code space, and the generator receives domain labels that can generate images for different domains. In a way, rather than considering each style as a separate domain, we can consider the whole set of images as a single large domain, with content and style being two different factors of variation, and our goal is to train an encoder-decoder that can disentangle these two factors. The idea of supplying domain information so that a single generator can generate images in different domains is also used in StarGAN (Choi et al., 2018). But in their work, there is no explicit content space, so it does not achieve the goal of disentangling style and content. Our view of style transfer being an instance of disentangled representation is also shared by (Huang et al., 2018), but their work considers mapping between two domains only.

**Disentangled Representation.** DC-IGN (Kulkarni et al., 2015) achieves clean disentanglement of different factor of variation in a set of images. However, the method requires very well-structured data. In particular, the method requires batches of training data with the same content but different style, as well as data with the same style but different content. For style transfer, it is often impossible to find many or even two images that depict the exact same content in different styles. On the other extreme, the method of Chen et al. Chen et al. (2016) is unsupervised and can discover disentangled factors of variations from unorganized data. Being unsupervised, there is no way to enforce the separation of a specific set of factors explicitly. The problem we are facing is in between, as we would like to enforce the meaning of the disentangled factors (style and content), but only one of the factors is controlled in the training data, as we can find images presumed to be in the same style that depict different content, but not vice versa. (Mathieu et al., 2016) is one example where the setting is the same to ours. Interestingly, related techniques can also be found in the field of audio processing. The problem of voice conversion, where an audio speech is mapped to the voice of a different speaker, has a similar structure to our problem. In particular, our approach is similar to (Chou et al., 2018).

## 3 Method

The training data must be organized by style. However, fine-grained labels of styles are difficult to obtain. Hence, we use the identity of artists, as a proxy for style. While an artist might have several styles and it might evolve over time, using an artist's identity as a proxy for style is a good approximation and an efficient choice, since the artist's label are readily available. As in (Chou et al., 2018), our method is divided into two stages.

**Stage 1: Style Independent Content Encoding.** In this stage, the goal is to train an encoder that encode as much information as possible about the content of an image, but no information about its style. We use per pixel L2 distance (not its square) for the reconstruction loss: for two (3-channel) images $X, Y \in \mathbb{R}^{h \times w \times 3}$, by $||X - Y||$ without any subscript we mean this distance:

$$||X - Y|| = \frac{1}{hw} \sum_{i=1}^{h} \sum_{j=1}^{w} ||X_{ij} - Y_{ij}||_2$$

We start from a solution that did not work well. Consider a simple encoder-decoder network with encoder $E(\cdot)$ and decoder $G(\cdot)$ whose sole purpose is to minimize the reconstruction loss:

$$\mathcal{L}_{\text{rec}} = \mathbb{E}_{x \sim p(x)} [||x - G(E(x))||] \qquad \min_{E,G} \mathcal{L}_{\text{rec}}$$

where $p(x)$ is the distribution of training samples. To prevent The encoder from encoding style information, we add an adversarial classifier $C(\cdot)$ that tries to classify the encoder's output by artist, while the encoder tries to maximize the classifier's loss:

$$\mathcal{L}_C = \mathbb{E}_{x,a \sim p(x,a)} [\text{NLL}(C(E(x)), a)] \qquad \min_{C} \mathcal{L}_C$$

$$\min_{E,G} \mathcal{L}_{\text{rec}} - \lambda \mathcal{L}_C \tag{1}$$

where $a$ is the integer ground truth label representing the author of image $x$ and $p(x,a)$ is the joint distribution of images and their authors. $\lambda$ is a weight factor. To resolve the conflicting goal that the generator needs style information to reconstruct the input but the encoder must not encode style information to avoid being successfully classified, we can learn a vector for each artist, representing their style. This "style code" is then provided to $G(\cdot, \cdot)$ which now takes two inputs. We introduce the style function $S(\cdot)$ which maps an artist to their style vector. Now the objective is

$$\mathcal{L}_{\text{rec}} = \mathbb{E}_{x,a \sim p(x,a)} [||x - G(E(x), S(a))||]$$

$$\min_{E,G,S} \mathcal{L}_{\text{rec}} - \lambda \mathcal{L}_C$$

Note that $S(\cdot)$ is not another encoding network. It does not see the input image, but only its artist label. This is essentially identical to the first stage of (Chou et al., 2018). In our experiments, we found that this method does not adequately prevent the encoder from encoding style information. We discuss this issue in appendix C.

We propose the following changes: instead of the code $E(x)$, $C(\cdot)$ tries to classify the generator's output $G(E(x), S(a'))$, which is the combination of the content of $x$ and the style of a *different* artist $a'$. In addition, akin to Variational Autoencoders (Kingma and Welling, 2013), the output of $E(\cdot)$ and $S(\cdot)$ are parameters of multivariate normal distributions and we introduce KL-divergence loss to constrain these distributions. To avoid the equations becoming too cumbersome, we overload the notation a bit so that when $E(\cdot)$ and $S(\cdot)$ are given to another network as input we implicitly sample a code from the distribution. The optimization objectives becomes

$$\mathcal{L}_C = \mathbb{E}_{\substack{x,a \sim p(x,a) \\ a' \sim p(a)}} [\text{NLL}(C(G(E(x), S(a'))), a)] \qquad \min_{C} \mathcal{L}_C$$

$$\mathcal{L}_{\text{rec}} = \mathbb{E}_{x,a \sim p(x,a)} [||x - G(E(x), S(a))||]$$

$$\mathcal{L}_{E\text{-KL}} = \mathbb{E}_{x \sim p(x)} [D_{\text{KL}}(E(x)||\mathcal{N}(0, I))] \qquad \mathcal{L}_{S\text{-KL}} = \mathbb{E}_{a \sim p(a)} [D_{\text{KL}}(S(a)||\mathcal{N}(0, I))]$$

$$\min_{E,G,S} \mathcal{L}_{\text{rec}} - \lambda_C \mathcal{L}_C + \lambda_{E\text{-KL}} \mathcal{L}_{E\text{-KL}} + \lambda_{S\text{-KL}} \mathcal{L}_{S\text{-KL}}$$

**Stage 2: Dual-Conditional Generator.** It is well known that autoencoders typically produce blurry outputs and fail to capture texture information which is an important part of style. To condition on styles, we base our approach on auxiliary classifier GANs (Odena et al., 2017). First, as usual, a discriminator $D(\cdot)$ tries to distinguish between real and generated samples, but instead of binary cross entropy, we follow (Mao et al., 2017) and use least squares loss. The discriminator's loss is

$$\mathcal{L}_{D\text{-real}} = \mathop{\mathbb{E}}_{x \sim p(x)} [(D(x) - 1)^2] \qquad \mathcal{L}_{D\text{-fake}} = \mathop{\mathbb{E}}_{\substack{x \sim p(x) \\ a' \sim p(a)}} [(D(G(E(x), S(a'))) + 1)^2]$$

while the generator's loss against the discriminator is

$$\mathcal{L}_{D\text{-adv}} = \mathop{\mathbb{E}}_{\substack{x \sim p(x) \\ a' \sim p(a)}} [D(G(E(x), S(a')))^2]$$

Here the encoder $E(\cdot)$, generator $G(\cdot, \cdot)$ and style function $S(\cdot)$ are inherited from stage 1. Note that in all the equations $a$ is sampled jointly with $x$ but $a'$ is independent. Then, similar to (Odena et al., 2017), a classifier is trained to classify training images to their correct authors:

$$\mathcal{L}_{C_2\text{-real}} = \mathop{\mathbb{E}}_{x,a \sim p(x,a)} [\text{NLL}(C_2(x), a)]$$

Unlike the generator and encoder, $C_2(\cdot)$ is a different classifier than the one in stage 1, thus we add subscript 2 to disambiguate, and shall refer to the stage 1 classifier as $C_1(\cdot)$. The generator try to generate samples that would be classified as the artist that it is conditioned on, by adding this to its loss function:

$$\mathcal{L}_{C_2\text{-adv}} = \mathop{\mathbb{E}}_{\substack{x \sim p(x) \\ a' \sim p(a)}} [\text{NLL}(C_2(G(E(x), S(a'))), a')] \tag{2}$$

But we differ from previous works on conditional GANs in the treatment of generated samples by the classifier. In (Odena et al., 2017) the classifier is cooperative in the sense that it would also try to minimize equation 2. In other works like (Choi et al., 2018; Chou et al., 2018; Mathieu et al., 2016) there is no special treatment for generated images, and the classifier does not optimize any loss function on them. In (Springenberg, 2015) the classifier is trained to be uncertain about the class of generated samples, by maximizing the entropy of $C_2(G(E(x), S(a')))$.

We train the classifier it to explicitly classify generated images conditioned on the style of $a$ as "not $a$". For this, we define what we call the "negative log-unlikelihood":

$$\text{NLU}(\mathbf{y}, i) = -\log(1 - y_i)$$

and take this as the classifier's loss on generated samples:

$$\mathcal{L}_{C_2\text{-fake}} = \mathop{\mathbb{E}}_{\substack{x \sim p(x) \\ a' \sim p(a)}} [\text{NLU}(C_2(G(E(x), S(a'))), a')]$$

We discuss the effect of an adversarial classifier in appendix C. While the discriminator and the classifier are commonly implemented as a single network with two outputs, we use separate networks for $C_2(\cdot)$ and $D(\cdot)$. To enforce the condition on content, we simply take $E$ and require that the generated samples be encoded back to its content input:

$$\mathcal{L}_{\text{cont}} = \mathop{\mathbb{E}}_{\substack{x \sim p(x) \\ a' \sim p(a)}} [||E(G(E(x), S(a'))) - E(x)||_2^2]$$

And our training objective for this stage is:

$$\min_{D} \mathcal{L}_{D\text{-real}} + \mathcal{L}_{D\text{-fake}} \qquad \min_{C_2} \mathcal{L}_{C_2\text{-real}} + \mathcal{L}_{C_2\text{-fake}}$$

$$\min_{G,S} \lambda_D \mathcal{L}_{D\text{-adv}} + \lambda_{C_2} \mathcal{L}_{C_2\text{-adv}} + \lambda_{\text{cont}} \mathcal{L}_{\text{cont}} + \lambda_{S\text{-KL}} \mathcal{L}_{S\text{-KL}}$$

Note that $E(\cdot)$ is fixed in stage 2. Since at the equilibrium $C_2(\cdot)$ is expected to classify real samples effectively, it would be good to pre-train $C_2(\cdot)$ on real samples only prior to stage 2. The training procedures are summarized in figure 1. KL-divergence omitted for clarity.

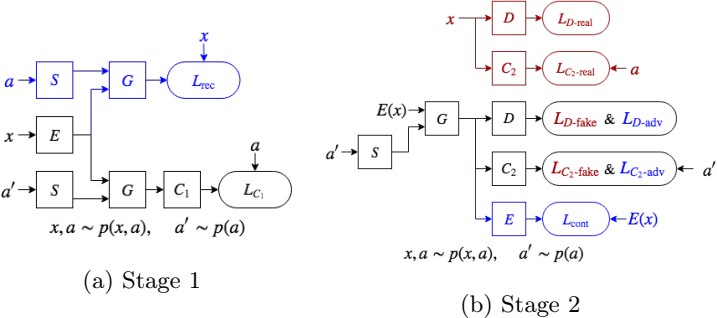

(a) Stage 1

(b) Stage 2

Figure 1: Training procedure. Squares are networks and rounded rectangles are loss terms. Blue parts are for $E$, $S$ and $G$, red parts are for $D$ and $C$. Black parts are common to both.

Table 1: Common part of network architecture

| Layer | - | SC | SC | SC | SC | SC | SC | F |
|---|---|---|---|---|---|---|---|---|
| Channel | 3 | 32 | 64 | 128 | 256 | 512 | 1024 | 2048 |
| Size | 256 | 128 | 64 | 32 | 16 | 8 | 4 | - |

## 4 EXPERIMENTS

For our main experiments presented here, we use anime illustrations obtained from Danbooru[1]. Further experiments on the NIST handwritten digit dataset can be found in appendix B, in which we also provide visualization and analysis of the encoder's output distribution for a better assessment of our method.

**Dataset.** We took all images with exactly one artist tag, and processed them with an open source tool AnimeFace 2009 (nagadomi, 2017) to detect faces. Each detected face is rotated into an upright position, then cropped out and scaled to $256 \times 256$ pixels. Every artist who has at least 50 image were used for training. Our final training set includes 106,814 images from 1139 artists.

**Network Architecture.** All networks are built from residue blocks (He et al., 2016). While the residue branch is commonly composed of two convolution layers, we only use one convolution per block and increase the number of blocks instead. ReLU activations are applied on the residue branch before it is added to the shortcut branch. For simplicity, all of our networks had an almost identical structure, with only minor differences in the input/output layers. The common part is given in table 1. The network is composed of the sequence of blocks in the first row (C = stride 1 convolution, S = stride 2 convolution, F = fully connected) with the number of output channels in the second row and spatial size of output feature map in the third row. For the generator, the sequence runs from right to left, while for other networks the sequence runs from left to right. On top of this common part, fully connected input/output layers are added for each network, with appropriate number of input/output features: for classifiers $C_1(\cdot)$ and $C_2(\cdot)$, the number of artists is 1,139; for the discriminator $D(\cdot)$, we use 1; for the encoder $E(\cdot)$, we use 2 parallel layers with 256 features for mean and standard deviation of output distribution; for the generator $G(\cdot, \cdot)$, the sum of the content and style dimensions is 512.

**Training.** We weight the loss terms with the hyperparameters in table 2, which also includes training parameters. $S(\cdot)$ is treated differently, since it is not a network but just matrices storing style codes. We use the same learning rate for every network in a single stage. In stage 2, we use RMSprop for our weight updating algorithm since using the

---

[1]danbooru.donmai.us

Table 2: Weighting and training hyperparameters

| Weight | Value |
|--------|-------|
| $\lambda_{C_1}$ | 0.2 |
| $\lambda_{E\text{-KL}}$ | $10^{-4}$ |
| $\lambda_{S\text{-KL}}$ | $2 \times 10^{-5}$ |
| $\lambda_D$ | 1 |
| $\lambda_{C_2}$ | 1 |
| $\lambda_{\text{cont}}$ | 0.05 |

| Stage | Learning rate $S$ | Learning rate Others | Algorithm | Batch | Time |
|-------|------|--------|-----------|-------|------|
| 1 | 0.005 | $5 \times 10^{-5}$ | Adam | 8 | 400k |
| $C_2$ pre-train | - | $10^{-4}$ | Adam | 16 | 200k |
| 2 | 0.01 | $2 \times 10^{-5}$ | RMSprop | 8 | 400k |

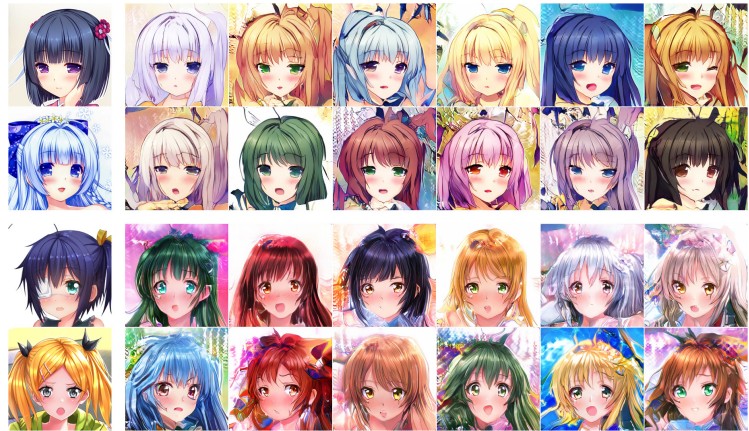

Figure 2: Images generated by fixing the style in each group of two rows and varying the content. Two different styles are shown. Leftmost column taken from training set, courtesy of respective artists. Top group: Sayori. Bottom group: Swordsouls.

momentum would sometimes cause instabilities in GANs. In every other stage we use the Adam optimizer. Timing is measured as number of iterations.

## 5 RESULTS

**Disentangled Representation of Style and Content.** To test our main goal of disentangling style and content in the generator's input code, we show that the style code and content code do indeed only control the respective aspect of the generated image, by fixing one and changing the other. In figure 2, in each group of two rows, the leftmost two images are examples of illustrations drawn by an artist in the dataset, and to the right are 12 samples generated from the style of the artist. Since this can be expected from a conventional class-conditional GAN, it is not the main contribution of our method. In particular, the strength of our method, is the ability to generate the same content in different styles where facial shapes, appearances, and aspect ratios are faithfully captured. We refer to Appendix A for additional results and discussions. Figure 3 shows 35 images generated with a fixed content and different style codes.

**Style Transfer.** As a straightforward application, we show some style transfer results in figure 4, with comparisons to existing methods, the original neural style transfer (Gatys et al., 2016) and StarGAN (Choi et al., 2018). We can see that neural style transfer seems to mostly apply the color of the style image to the content image, which, by our definition, not only fails to capture the style of the style image but also alters the content of the content image. StarGAN managed to transfer the overall use of color of the target artist and some other prominent features like size of the eyes, but otherwise fails to capture intricate style elements. Our method transfers the style of the target artist much more faithfully.

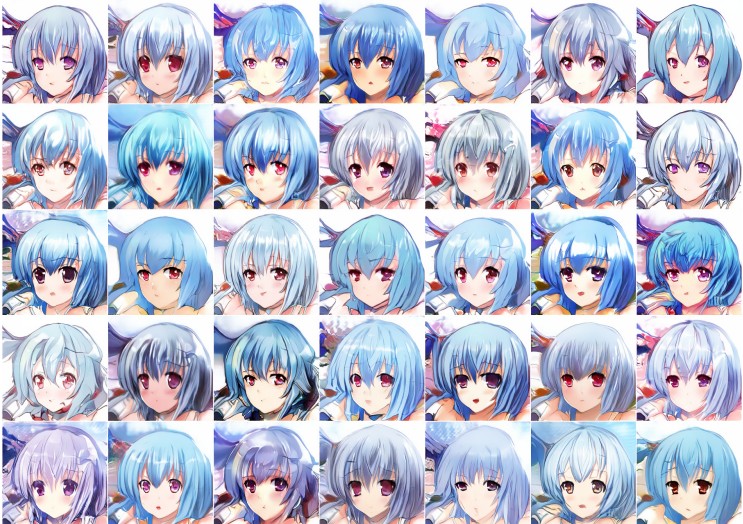

Figure 3: Images generated from a single content code and an assortment of styles. Including both style of artists from the training set and style codes randomly samples from the style distribution

**Evaluation.** Other than visual results, there is no well-established quantitative measure for the quality of style transfer methods and accessing experts for anime styles is challenging for a proper user study. The evaluation in (Chou et al., 2018) is audio specific. In (Choi et al., 2018), the classification accuracy on the generated samples is given as a quality measure. However, this seemingly reasonable measure, as we shall argue in appendix C.2, is not adequate. Nevertheless, we do report that the samples generated by our generator can be classified by style by our classifier with 86.65% top-1 accuracy. Detailed testing procedure, results and analysis can be found in appendix C.2. A comprehensive ablation study is given in Appendix C, where major differences from previous approaches are evaluated. The effectiveness of stage 1, the disentangling encoder, is evaluated quantitatively in section B.4.

**Limitations.** Due to the relative scarcity of large labelled collection of artworks depicting a consist set of subjects, we have thus far not collected and tested our method on examples beyond portraits and artistic styles beyond anime illustrations. We also noticed some inconsistencies in small features, such as eye colors, presumably due to small features being disfavored by the per-pixel reconstruction loss in stage 1, as well as our choice of architecture with fixed-size code rather than fully convolutional. Additional discussions can be found in Appendix A.

## 6 CONCLUSION

We introduced a Generative Adversarial Disentangling Network which enables true semantic-level artwork synthesis using a single generator. Our evaluations and ablation study indicate that style and content can be disentangled effectively through our a two-stage framework, where first a style independent content encoder is trained and then, a content and style-conditional GANs is used for synthesis. While we believe that our approach can be extended to a wider range of artistic styles, we have validated our technique on various styles within the context of anime illustrations. In particular, this techniques is applicable, as long as we disentangle two factors of variation in a dataset and only one of the factors is labelled and controlled. Compared to existing methods for style transfer, we show significant improvements in terms of modeling high-level artistic semantics and visual quality. In the future, we hope to extend our method to styles beyond anime artworks, and we are also interested in learning to model entire character bodies, or even entire scenes.

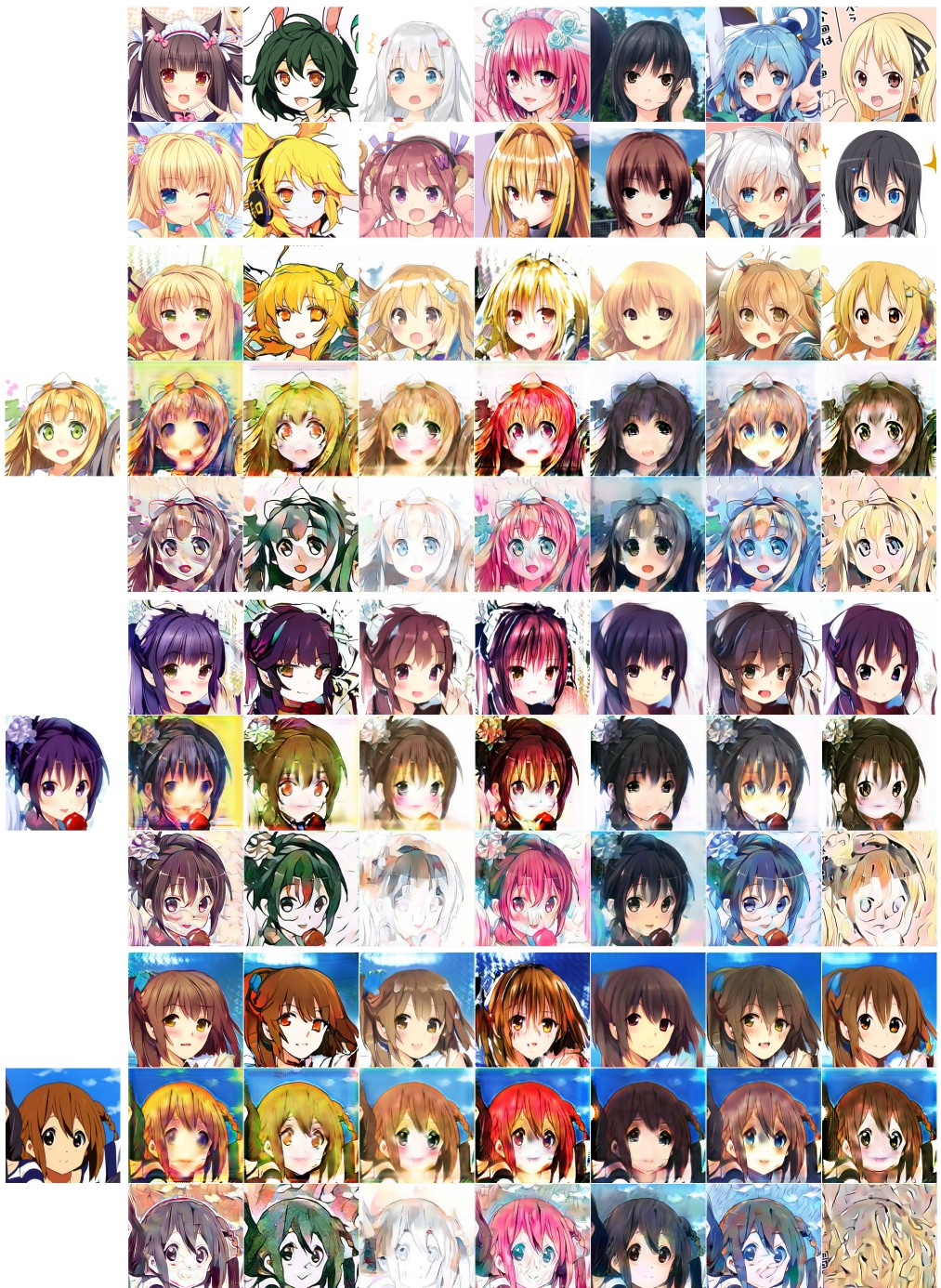

Figure 4: Style transfer results. In the top two rows, in each column are two samples from the training dataset by the same artist. In each subsequent group of three rows, the leftmost image is from the training dataset. The images to the right are style transfer results generated by three different methods, from the content of the left image in the group and from the style of the top artist in the column. In each group, first row is our method, second row is StarGAN and third row is neural style. For neural style, the style image is the topmost image in the column. Training samples courtesy of respective artists. Style samples, from left: Sayori, Ideolo, Peko, Yabuki Kentarou, Coffee-kizoku, Mishima Kurone, Ragho no Erika. Content samples, from top: Kantoku, Koi, Horiguchi Yukiko.

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

## A  Additional Discussions

### A.1  Thoughts on Style and Style Transfer

While works on this topic has been numerous, we feel that one fundamental question is not often carefully addressed: what is style, in a deep learning setting?

As stated in (Gatys et al., 2016), which is based on an earlier work on neural texture synthesis (Gatys et al., 2015), the justification for using Gram matrices of neural network features as a representation of style is that it captures statistical texture information. So, in essence, "style" defined as such is a term for "texture statistics", and the style transfer is limited to texture statistics transfer. Admittedly, it does it in smart ways, as in a sense the content features are implicitly used for selecting which part of the style image to copy the texture statistics from.

As discussed in section 2 above, we feel that there is more about style than just feature statistics. Consider for example the case of caricatures. The most important aspects of the style would be what facial features of the subjects are exaggerated and how they are exaggerated. Since these deformations could span a long spatial distance, they cannot be captured by local texture statistics alone.

Another problem is domain dependency. Consider the problem of transferring or preserving color in style transfer. If we have a landscape photograph taken during the day and want to change it to night by transferring the style from another photo taken during the night, or if we want to change the season from spring to autumn, then color would be part of the style we want to transfer. But if we have a still photograph and want to make it an oil painting, then color is part of the content, we may want only the quality of the strokes of the artwork but keep the colors of our original photo.

People are aware of this problem and in (Gatys et al., 2017), two methods, luminance-only style transfer and color histogram matching, are developed to optionally keep the color of the content image. However, color is only one aspect of the image for which counting it as style vs. content could be an ambiguity. For more complicated aspects, the option to keep or to transfer may not be easily available.

We make two observations here. First, style must be more than just feature statistics. Second, the concept of "style" is inherently domain-dependent. In our opinion, "style" means different ways of presenting the same subject. In each different domain, the set of possible subjects is different and so is the set of possible ways to present them.

So, we think that any successful style transfer method must be adaptive to the intended domain and the training procedure must actively use labelled style information. Simple feature based methods will never work in the general setting. This includes previous approaches which explicitly claimed to disentangle style and content, such as in (Kazemi et al., 2019) which adopts the method in the original neural style transfer for style and content losses, and also some highly accomplished methods like (Liao et al., 2017).

As a side note, for these reasons we feel that some previous methods made questionable claims about style. In particular, works like (Huang et al., 2018) and StyleGAN (Karras et al., 2018) made reference to style while only being experimented on collections of real photographs. By our definition, in such dataset, without a careful definition and justification there is only one possibly style, that is, photorealistic, so the distinction between style and content does not make sense, and calling a certain subset of factors "content" and others "style" could be an arbitrary decision.

This is also why we elect to not test our method on more established GAN datasets like CelebA or LSUN, which are mostly collections of real photos.

### A.2  More on Results

The differences in style can be subtle, and readers may not be familiar enough with anime illustrations to be able to recognize them. We hint at several aspects to which the reader

may want to pay attention: overall saturation and contrast, prominence of border lines, overall method of shading (flat vs. 3-dimensional), size and aspect ratio of the eyes, shape of the irides (round vs. angular), shininess of the irides, curvature of the upper eyelids, height of lateral angle of the eyes, aspect ratio of the face, shape of the chin (round vs. angular), amount of blush, granularity of hair strands, prominence of hair shadow on forehead, treatment of hair highlight (intensity, no specularities / dots along a line / thin line / narrow band / wide band, clear smooth boundary / jagged boundary / fuzzy boundary, etc.).

If we specifically examine these style elements in the style transfer results in figure 4, it should be evident that our method has done a much better job. That being said, our network is not especially good at preserving fine content details.

This can be said to be a result from our choice of architecture: we use an encoder-decoder with a fixed-length content code while StarGAN uses a fully convolutional generator with only 2 down-sampling layers, which is able to preserve much richer information about the content image.

Part of the reason is that we consider it a desirable feature to be able to sample from the content distribution. In convolutional feature maps, features at different locations are typically correlated in complicated ways which makes sampling from the content distribution impractical. Unsurprisingly previous works adopting fully convolutional networks did not demonstrate random sampling of content, which we did.

Another reason is that we feel that the ability of fully convolutional networks to preserve content has a downside: it seems to have a tendency to preserve image structures down to the pixel level. As can be observed in figure 4, StarGAN almost exactly preserves location of border lines between different color regions. As we have mentioned, part of the artists' style is different shape of facial features, and a fully convolutional network struggle to capture this. We suspect that for a fully convolutional architecture to work properly, some form of non-rigid spatial transformation might be necessary. This can be one of our future directions.

### A.3    Inconsistency of Small Features

As can be seen from many of the visualizations, the color of the eyes, and facial expression - which largely depends on the shape of the mouth - are sometimes not well preserved when changing the style codes. We thought that this could be the result of the choice of the reconstruction loss function in stage 1. The loss is averaged across all image pixels, which means the importance of a feature to the network is proportional to its area. With the code length being limited, the network prioritizes encoding large features, which can be seen from the fact that large color regions in the background are more consistent than eyes and mouth.

For this particular problem, we can utilize additional tags: almost all images gathered from Danbooru are tagged according to eye color and facial expressions. We could train the encoder to classify for these tags while the generator could be conditioned on these tags. For general problems however, we may need loss functions that are more aligned with humans' perception of visual importance.

Another observation is that sometimes in a generated image the two eyes are of different color. We point out that the dataset do contain such images, and while still rare, the prevalence of heterochromia in anime characters is much higher than in real life. We think that this could be enough to force the encoder to use two different set of dimensions to encode the color of each eye, thus causing random samples to have different colored eyes.

## B    Experiments on NIST Dataset

We conduct further experiments on the recently released full NIST handwritten digit dataset (Yadav and Bottou, 2019), which is a superset of the MNIST dataset. In addition to having many more images (a total of 402,953 images), the NIST dataset comes with more metadata, among which the identity of the writer in particular is useful for our study (3579 different writers). We show the ability of our network to disentangle between digit class and writer

Table 3: Weighting and training hyperparameters for NIST

| Parameter | $\mathcal{W}$ vs. $\mathcal{D} + \mathcal{R}$ | $\mathcal{D}$ vs. $\mathcal{W} + \mathcal{R}$ |
|---|---|---|
| $\lambda_{C_1}$ | 0.1 | 0.1 |
| $\lambda_{E\text{-KL}}$ | $10^{-4}$ | $10^{-4}$ |
| $\lambda_{S\text{-KL}}$ | $10^{-4}$ | $10^{-4}$ |
| $\lambda_D$ | 1 | 1 |
| $\lambda_{C_2}$ | 0.2 | 1 |
| $\lambda_{\text{cont}}$ | 0.5 | 0.1 |
| Stage 1 time | 300k | 300k |
| $C_2$ pre-train time | 800k | 100k |
| Stage 2 time | 320k | 320k |

identity when only the writer label is known but not the digit label, or vice versa, when only the digit label is known but not the writer label.

Strictly speaking, we are not precisely disentangling between digit class and writer identity: as we will show shortly, non-negligible variation exists when the same person is writing the same digit, so there are actually three set of factors that determines the appearance of a written digit: digit class (denoted $\mathcal{D}$), writer identity ($\mathcal{W}$) and the rest ($\mathcal{R}$). We have labels for $\mathcal{D}$ and $\mathcal{W}$ in the dataset. Our purpose is to disentangle between labelled variations and unlabelled variations. So, with the label for digit and writer available, we show the following two experiments: disentangling between $\mathcal{D}$ and $\mathcal{W} + \mathcal{R}$ when only digit label is used for training, and disentangling between $\mathcal{W}$ and $\mathcal{D} + \mathcal{R}$ when only writer label is used for training.

### B.1 Setup

The images of handwritten digits are of size $28 \times 28$. Accordingly, the size of our network was reduced to 3 levels with two convolution residue blocks each, with 64, 128 and 256 channels.

The code length for digit class, writer identity and the rest of features were 16, 128 and 32 respectively. Thus, for $\mathcal{W}$ vs. $\mathcal{D} + \mathcal{R}$ the labelled feature had 128 dimensions and the unlabelled feature had $16 + 32 = 48$ dimensions, while for $\mathcal{D}$ vs. $\mathcal{W} + \mathcal{R}$ the labelled feature had 16 dimensions and the unlabelled feature had $128 + 32 = 160$ dimensions.

Weighting parameters and training times were also adjusted and are given in table 3.

### B.2 Disentangling by Writer Label

First we show the scenario where the label for writer identity is known, similar to the main experiment where the artist is known. Generated samples of all digits and randomly selected writers are shown in figure 5. Samples in the same column is from the same writer, whose code is that learned by $S(\cdot)$, while samples in the same row has the same $\mathcal{D} + \mathcal{R}$ code, which is obtained by taking the average of $E(x)$ from all images $x$ in the same digit class.

Although we fell that it is clear that we can control style and content independently, given the fact that the perception of style is subjective, here we demonstrate how cleanly $\mathcal{W}$ has been disentangled from $\mathcal{D} + \mathcal{R}$ by visualizing the output distribution of $E(\cdot)$.

Principal Component Analysis is commonly used for projecting a high dimensional feature space to a low dimensional one. But considering that feature scaling can easily skew the result, and because in section C.1 we will be visualizing the change of the output distribution of $E(\cdot)$ during training which require the same projection be used for different distributions, we chose to use Nested Dropout (Rippel et al., 2014) so that code dimensions are automatically learned and ordered by importance and we can simply project to the first two dimensions for visualization.

Figure 6a shows the distribution of $E(x)$ over the whole dataset projected to the first two dimensions, with color representing digit class. In figure 6b is the output of a vanilla VAE

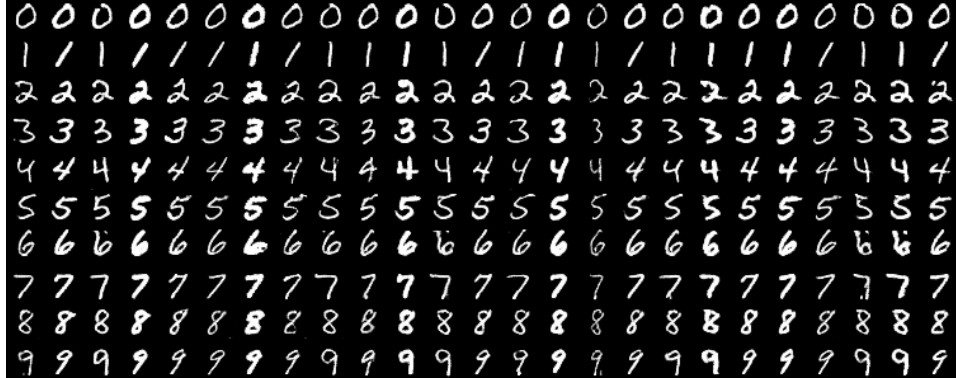

Figure 5: Generated samples when disentangling $\mathcal{W}$ from $\mathcal{D} + \mathcal{R}$

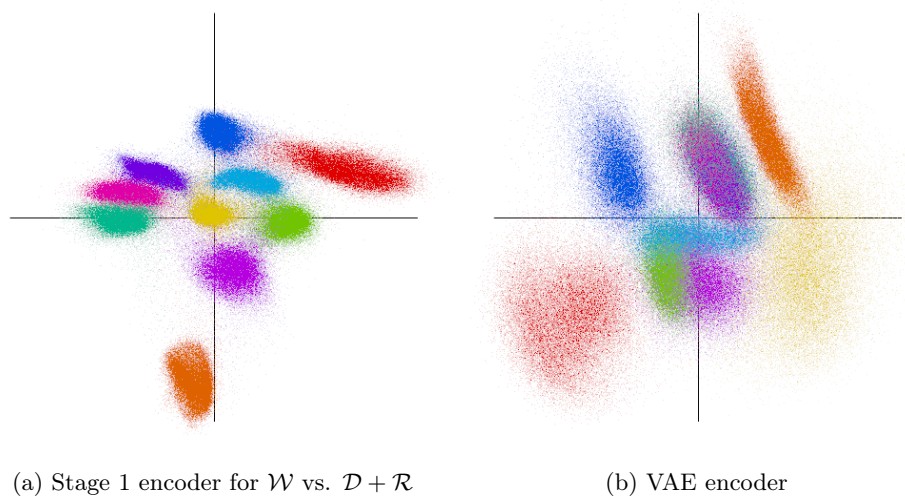

(a) Stage 1 encoder for $\mathcal{W}$ vs. $\mathcal{D} + \mathcal{R}$        (b) VAE encoder

Figure 6: Comparison of output distribution between stage 1 encoder and vanilla VAE.

with the same structure trained on the same dataset. Comparing to a vanilla VAE, our $E(\cdot)$ produced a distribution in which images depicting the same digit are more closely clustered together while images depicting different digits are more cleanly separated. In particular, we obtained 10 clearly distinct clusters with only a two dimensional feature space, while with VAE three of the digits are almost completely confused (they needed a third dimension to be separated). This is because we have purged information about the writer from the encoder: Since the identity of the writer is independent from digit class, by removing writer information, variation within the same digit class were reduced. Remember that we are able to achieve this clustering effect without knowing digit label during training.

### B.3  DISENTANGLING BY DIGIT LABEL

Our method is not limited to the case where the labelled feature correspond to style while the unlabelled feature correspond to content. In fact, as we will show here, we can achieve the direct opposite, with the labelled feature corresponding to content and the unlabelled feature corresponding to style: we disentangle $\mathcal{D}$ from $\mathcal{W} + \mathcal{R}$. Generated samples of all digits and randomly selected writers are shown in figure 7. Samples in the same row represents the same digit, whose code is that learned by $S(\cdot)$, while samples in the same column has the same $\mathcal{W} + \mathcal{R}$ code, which is obtained by taking the average of $E(x)$ from all images $x$ by the same writer.

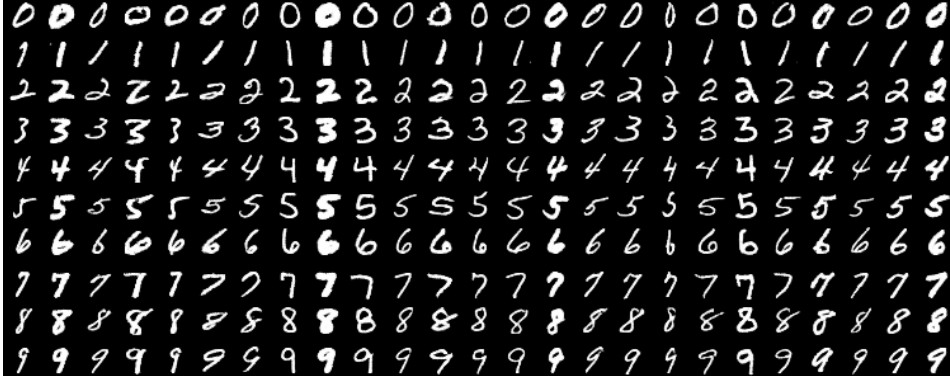

Figure 7: Generated samples when disentangling $\mathcal{D}$ from $\mathcal{W} + \mathcal{R}$

Figure 8: Variation within the same digit written by the same writer

We can observe that the variation in the same row is much more dramatic in figure 7 than in figure 5, despite that in both cases we can (roughly) consider each row to have the same content and each column to have the same style. This is in fact the correct behaviour: in figure 5 the variation in each row is due to $\mathcal{W}$ only while in figure 7 the variation in each row is due to the combined effect of $\mathcal{W}$ and $\mathcal{R}$. This reveals the fact that, even within multiple images of the same digit written by the same person, the variation can be significant.

We can verify this in figure 8: in the left 5 columns are samples by the same writer taken from the training dataset. In the right 10 columns are samples generated by the generator in section B.2. The writer code is that of the same writer as the samples to the left, as learned by $S(\cdot)$, while the $\mathcal{D} + \mathcal{R}$ codes are obtained by adding some noise to the average $E(x)$ of different digit classes. The training data shows certain amount of variation within the same digit written by the same writer, and we can achieve similar variations by fixing $\mathcal{W}$ and $\mathcal{D}$ and only changing $\mathcal{R}$.

Similar to section B.2, we look at the distribution of $E(x)$ again. This time, since the digit class is the labeled feature which we want to purge from the encoder, if we were successful, the distributions of each digit should be indistinguishable. The distribution of each individual digit are given in figure 9. These distributions are indeed very similar. Compare this to figure 6b.

B.4 Quantitative Analysis of Disentangling on NIST Dataset

In sections B.2 and B.3 we gave qualitative evaluation of the full pipeline by various visualizations. In this section we focus on the stage 1 encoder and give some quantitative evaluation of the effect of disentanglement.

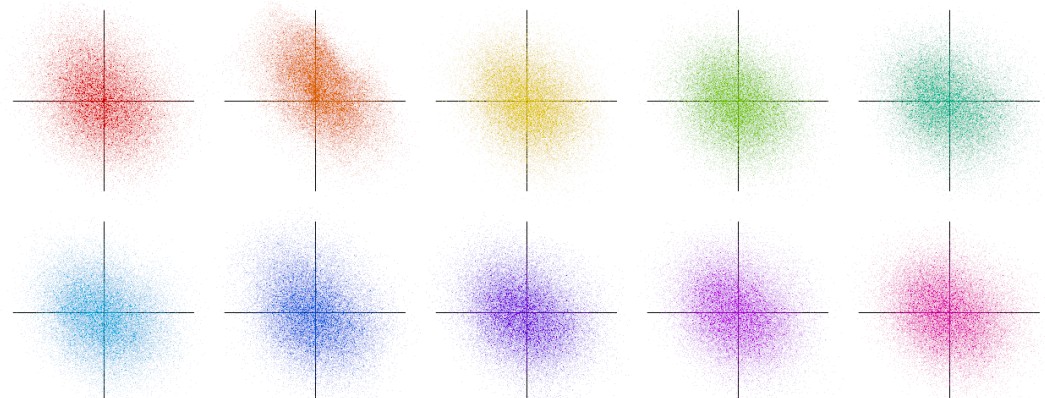

Figure 9: Distribution of each digit from stage 1 encoder for $\mathcal{D}$ vs. $\mathcal{W} + \mathcal{R}$

Table 4: Mean Euclidean distance of a sample to the center of its class

| Encoder | By writer | By digit | Whole dataset |
|---|---|---|---|
| $E_{\mathcal{W}}$ | 1.2487 | 0.2788 | 1.2505 |
| $E_{\mathcal{D}}$ | 0.7929 | 1.2558 | 1.2597 |
| $E_{\mathcal{V}}$ | 1.2185 | 0.4672 | 1.2475 |

(a) First 2 dimensions

| Encoder | By writer | By digit | Whole dataset |
|---|---|---|---|
| $E_{\mathcal{W}}$ | 2.6757 | 2.0670 | 2.6957 |
| $E_{\mathcal{D}}$ | 2.4020 | 2.6699 | 2.7409 |
| $E_{\mathcal{V}}$ | 2.6377 | 1.7629 | 2.7363 |

(b) First 8 dimensions

The encoder aims to encode the unlabelled feature in the dataset while avoiding encoding the labelled feature. Intuitively, this means that each unlabelled class should be distributed differently in the encoder's code space while each labelled class should be distributed similarly. We make this intuition precise by considering two metrics: average Euclidean distance of a sample in the code space to the mean of its class, and the performance of a naive Bayesian classifier using the code as feature.

First, consider the average distance metric. For the disentangling encoder trained on writer label (denoted $E_{\mathcal{W}}$ in this section), if it does successfully encode digit information but not writer information, samples of the same digit should be clustered more tightly together than the whole dataset, while samples by the same writer should not be clustered noticeably more tightly than the whole dataset. In addition, samples of the same digit should be clustered more tightly together than for a VAE encoder not trained to disentangle ($E_{\mathcal{V}}$), and samples by the same writer should be clustered less tightly. For the disentangling encoder trained on digit label ($E_{\mathcal{D}}$), the opposite should be true. So, for each of $E_{\mathcal{W}}$, $E_{\mathcal{V}}$ and $E_{\mathcal{D}}$, we compute the average Euclidean distance of a sample to the center of its class, with the classification being by writer, by digit, or just the whole as a single class.

We fond that the most distinctive information are all encodes in the first few dimensions of the code space, with later dimensions being more random, thus disturbing the clustering effect rather than helping. So we truncate the encoder's output to either two dimension or eight dimensions, and give the result for both. The result is shown in table 4. In each row, the red number should be noticeably smaller than the black number while the blue number should be similar to the black number. In each column, the red number should be noticeably smaller than the black number, while the blue number would preferably be larger.

Table 5: Average probability given to the correct class

| Encoder | By writer | By digit |
| --- | --- | --- |
| $E_{\mathcal{W}}$ | 0.000293 | 0.9001 |
| $E_{\mathcal{D}}$ | 0.001441 | 0.1038 |
| $E_{\mathcal{V}}$ | 0.000337 | 0.6179 |

(a) First 2 dimensions

| Encoder | By writer | By digit |
| --- | --- | --- |
| $E_{\mathcal{W}}$ | 0.000363 | 0.9327 |
| $E_{\mathcal{D}}$ | 0.002845 | 0.1015 |
| $E_{\mathcal{V}}$ | 0.000843 | 0.9380 |

(b) First 8 dimensions

Table 6: Average rank of the correct class

| Encoder | By writer | By digit |
| --- | --- | --- |
| $E_{\mathcal{W}}$ | 1608 | 1.12 |
| $E_{\mathcal{D}}$ | 582 | 5.20 |
| $E_{\mathcal{V}}$ | 1409 | 1.49 |

(a) First 2 dimensions

| Encoder | By writer | By digit |
| --- | --- | --- |
| $E_{\mathcal{W}}$ | 1330 | 1.12 |
| $E_{\mathcal{D}}$ | 422 | 3.98 |
| $E_{\mathcal{V}}$ | 838 | 1.08 |

(b) First 8 dimensions

We can see that $E_{\mathcal{W}}$ causes the samples to cluster very tightly by digit while $E_{\mathcal{D}}$ has the opposite effect, which is consistent with the visualization in figures 6a and 9. Conversely, $E_{\mathcal{D}}$ causes the sample to cluster somewhat more tightly by writer, while $E_{\mathcal{W}}$ has the opposite effect.

Second, we study the classification performance in the encoder's output space. For this purpose, we use a naive Bayesian classifier with each class modelled by an axis-aligned multivariate normal distribution. That is, for each class we take all samples of that class, fit a normal distribution with the covariance matrix being a diagonal matrix, then for each sample compute the likelihood of it being generated by each class and give a class distribution by Bayes' theorem. For each of $E_{\mathcal{W}}$, $E_{\mathcal{V}}$ and $E_{\mathcal{D}}$ we classify by writer and by digit, with first two dimensions and then with first 8 dimensions of the code space.

The average probability assigned to the correct class, average rank of the correct class and top-1 classification accuracy are given in tables 5, 6 and 7 respectively. Similar to the average distance metric, in each table the red number should be better than the black number in the same column while the blue number should be worse, with "better" meaning high average probability to the correct class, lower average rank and higher top-1 accuracy. A lot of observations can be made from these numbers. For example, with the first two features produced by $E_{\mathcal{W}}$ digits can be classified with a top-1 accuracy of 94%, while with the first two features produced by $E_{\mathcal{D}}$ the top-1 accuracy is only 13%, just a bit better than random guess.

## C ABLATION STUDY

We provide additional details on our method in Section 3, along with ablation studies to demonstrate their effectiveness.

Table 7: Top-1 accuracy

| Encoder | By writer | By digit |
| --- | --- | --- |
| $E_{\mathcal{W}}$ | 0.000454 | 0.94 |
| $E_{\mathcal{D}}$ | 0.005331 | 0.13 |
| $E_{\mathcal{V}}$ | 0.000846 | 0.70 |

(a) First 2 dimensions

| Encoder | By writer | By digit |
| --- | --- | --- |
| $E_{\mathcal{W}}$ | 0.001400 | 0.94 |
| $E_{\mathcal{D}}$ | 0.015424 | 0.23 |
| $E_{\mathcal{V}}$ | 0.004946 | 0.95 |

(b) First 8 dimensions

## C.1 Using Reconstruction Result as Stage 1 Classifier Input

By design, assuming sufficiently powerful networks, the stage 1 method in (Chou et al., 2018) should reconstruct $x$ perfectly with $G(E(x), S(a))$ while at the same time not include any style information in $E(x)$. But in reality, this method worked poorly in our experiments. We discuss our conjecture here.

We think the problem is that the distribution of $E(x)$ is unconstrained and the encoder-decoder can exploit this to trick the classifier while still encode style information in $E(x)$. Assume that the last layer weight of $E(\cdot)$ is $W_1 \in \mathbb{R}^{m \times d}$ and the first layer weight of $G(\cdot, \cdot)$ (connected to its content input) is $W_2 \in \mathbb{R}^{d \times n}$ where $d$ is the length of the content code and $n$ and $m$ are the number of features in relevant layers. Then for any invertible matrix $H \in \mathbb{R}^{d \times d}$, replacing $W_1$ with $W_1 H$ and $W_2$ with $H^{-1} W_2$ would give us an autoencoder that computes the exact same function as before but with a linearly transformed distribution in the code space. Note that such a "coordination" is not limited to be between the innermost layers, and the parameters of $E(\cdot)$ and $G(\cdot, \cdot)$ may be altered to keep the reconstruction almost unchanged but at the same time transform the code distribution in complicated ways beyond simple linear transformation.

As a result, the encoder-decoder can transform the code distribution constantly during training. The classifier would see a different input distribution in each iteration and thus fail to learn to infer the artist from style information that could possibly be included in the output of $E(\cdot)$, and thus would be ineffective at preventing $E(\cdot)$ from encoding style information.

Note that constraining the code distribution of the whole training set does not help. For example, if we constraint the code distribution to be standard normal as in VAE, we can take $H$ in the previous example to be an orthonormal matrix. The transformed distribution on the whole dataset would still be standard normal but the distribution of artworks by each different artist would change.

We noticed that this problem would be alleviated if instead of the code $E(x)$, $C_1(\cdot)$ tries to classify the reconstruction result $G(E(x), S(a))$ since the output distribution of the decoder is "anchored" to the distribution of training images and cannot be freely transformed. But the output of $G(\cdot, \cdot)$, being an reconstruction of the input image, will inevitably contain some style information, so giving $G(E(x), S(a))$ as input to $C_1(\cdot)$ will again cause the problem of conflicting goals between reconstruction and not encoding style information. The important observation is that, the input given to $C_1(\cdot)$ is allowed to contain style information as long as it is not the style of the real author of $x$! Thus we arrived at our proposed method where the classifier sees $G(E(x), S(a'))$, the image generated by the encoder's output on $x$ and the style of an artist $a'$ different than the author of $x$.

Strictly speaking, by classifying the output of the generator we can only ensure that $E(\cdot)$ and $G(\cdot, \cdot)$ would jointly not retain the style of $x$ in $G(E(x), S(a'))$. $E(\cdot)$ may still encode style information which is subsequently ignored by $G(\cdot, \cdot)$. To solve this, we constrain the output distribution of $E(\cdot)$ with a KL-divergence loss as in VAE. With such a constraint, $E(\cdot)$ would be discouraged from encoding information not used by $G(\cdot, \cdot)$ since that would not improve the reconstruction but only increase the KL-divergence penalty.

To compare the effect of classifiers seeing different inputs, we train an alternative version of stage 1 using the method in (Chou et al., 2018), where instead of a convolutional network that gets the generator's output, we use a multi-layer perceptron that gets the encoder's output as the classifier. We use an MLP with 10 hidden layers, each having 2048 features. We then compare $G(E(x), \mathbf{0})$ with $G(E(x), S(a))$, that is, image generated with the content code of an image $x$ plus an all-zero style code versus the correct style code. A successful stage 1 encoder-decoder should reconstruct the input image faithfully with the correct style code but generate style-nutral images with the zero style code. We also give the reconstruction of an conventional VAE for reference. The result is shown in figure 10.

As autoencoders are in general bad at capturing details, the readers may want to pay attention to more salient features like size of the eyes, shape of facial contour and amount of blush. We can see that these traits are clearly preserved in the style-neutral reconstruction

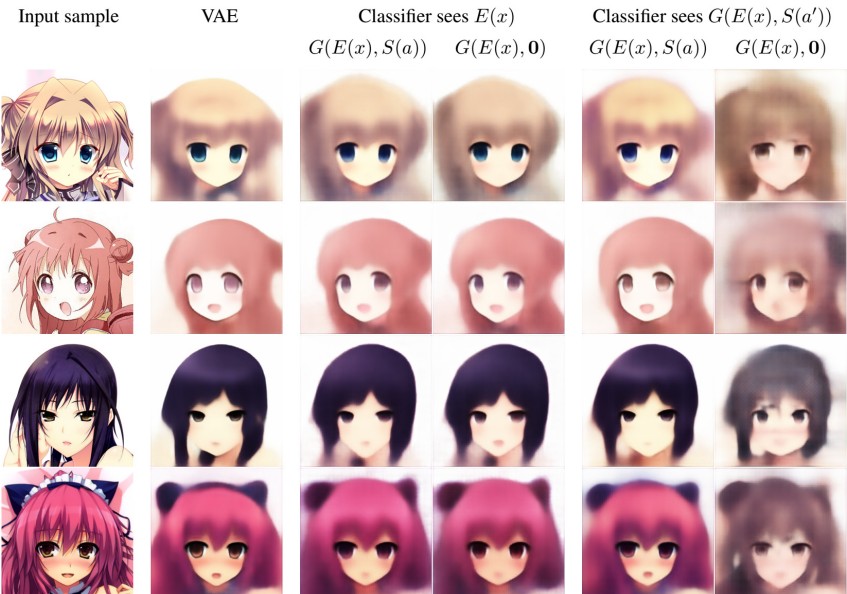

Figure 10: Comparison of stage 1 image reconstruction with correct style and zero style, using different methods. Column 1: images from the dataset. Column 2: VAE reconstruction. Column 3: MLP classifier, correct style. Column 4: MLP classifier, zero style. Column 5: Our classifier, correct style. Column 6: Our classifier, zero style. Training images courtesy of respective artists. From top: Azumi Kazuki, Namori, Iizuki Tasuku, Tomose Shunsaku.

when an MLP is used for classifying $E(x)$. Our method, while not able to completely deprive the output of style information from the input in the style-neutral reconstruction and also altered the content by a bit, performed much better, and did so without compromising the reconstruction with the correct style.

We point out that this is not due to the MLP not having enough capacity to classify $E(x)$: when the encoder is trained to only optimize for reconstruction and ignore the classifier by setting $\lambda_{C_1} = 0$, the classifier is able to classify $E(x)$ with higher than 99% top-1 accuracy.

As a direct evidence in support of our conjecture that the performance of stage 1 with MLP classifier was hurt due to instabilities of the encoder's output distribution, we look at the network trained on NIST with writer label in section B.2 again and track the change of the encoder's output distribution. For comparison, we repeat the training but substitute the classifier for a 6-layer MLP. As a reference, we also track the output distribution of a vanilla VAE. The result is shown in figure 11. The encoders' distribution were visualized from iteration 200,000 to iteration 300,000 at an interval of 20,000 iterations.

The encoder's distribution when an MLP classifier is used was clearly unstable. This is not an intrinsic property of autoencoders, as the output distribution of the vanilla VAE encoder remained essentially unchanged, so the instability must have been introduced by the adversarial training against the classifier. In contrast, the output distribution of our stage 1 encoder was much more stable. Although the distribution did change, each digit class only fluctuated within a small range and the relative position of the classes remained consistent.

We can also see that although the encoder trained with MLP classifier is still better at clustering the same digit than VAE, it did not separate between different digits very clearly.

## C.2 Adversarial Stage 2 Classifier

One important difference in our style-conditional GAN training compared to previous class-conditional GAN approaches is that, in our case, not only is the discriminator adversarial

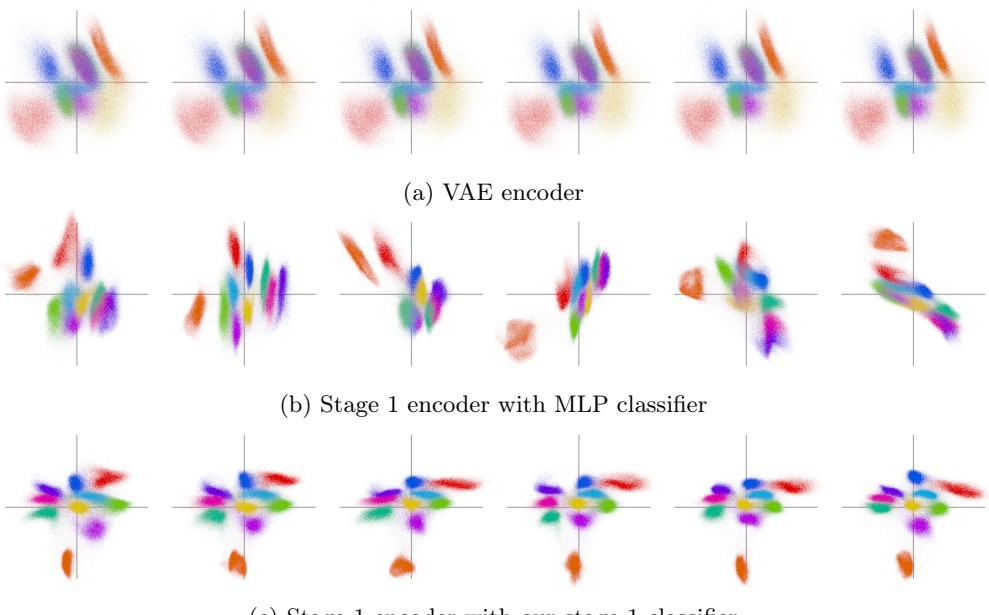

(a) VAE encoder

(b) Stage 1 encoder with MLP classifier

(c) Stage 1 encoder with our stage 1 classifier

Figure 11: Change of output distribution of different encoders

but the classifier as well. The classifier explicitly tries to classify samples generated using an artist $a$ as "not by $a$".

The rationale is that, by just correctly classifying the real samples, the classifier may not be able to learn all the aspects about an artist's style. Imagine the hypothetical scenario where each artist draws the hair as well as the eyes differently. During training, the classifier arrives at a state where it can classify the real samples perfectly by just considering the hair and disregarding the eyes. There will then be no incentive for the classifier to learn about the style of the eyes since that will not improve its performance.

If the classifier is adversarial, this will not happen: as long as the generator has not learned to generate the eyes in the correct style, the classifier can improve by learning to distinguish the generated samples from real ones by the style of their eyes.

In general, a non-adversarial classifier only needs to learn as much as necessary to tell different artists apart, while an adversarial classifier must understand an artist's style comprehensively.

To study the effect of this proposed change, we repeat stage 2 but without adding $\mathcal{L}_{C_2\text{-fake}}$ to the classifier's loss.

In figure 12 we generate samples using this generator trained without an adversarial classifier, from the exact same content code and artists as in figure 2. While we leave the judgment to the readers, we do feel that making the classifier adversarial does improve the likeness.

The choice of "negative log-unlikelihood" as the classifier's loss on generated samples might seem a bit unconventional:

$$\text{NLU}(\mathbf{y}, i) = -\log(1 - y_i)$$
$$\mathcal{L}_{C_2\text{-fake}} = \underset{\substack{x \sim p(x) \\ a' \sim p(a)}}{\mathbb{E}} \left[ \text{NLU}(C_2(G(E(x), S(a'))), a') \right]$$

To many, maximizing the negative log-likelihood would be a much more natural choice. The major concern here is that the negative log-likelihood is not upper-bounded. Suppose that $C_2(\cdot)$ is trained using such an objective. There would then be little to be gained by trying to classify real samples correctly because $\mathcal{L}_{C_2\text{-real}}$ is lower-bounded. To decrease its

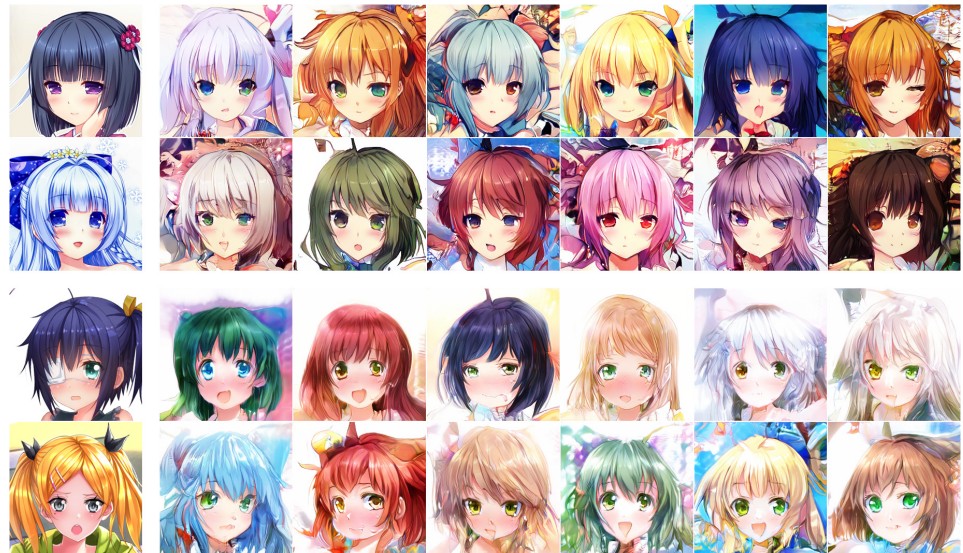

Figure 12: Images generated from fixed style and different contents, when stage 2 classifier is not adversarial.

loss, it would be much more effective to increase the negative log-likelihood on generated samples. The result is that the $C_2(\cdot)$ simply ignores the loss on real samples while the negative log-likelihood on generated samples quickly explodes. If a classifier cannot classify the real samples correctly, one would not expect the generator to learn the correct styles by training it against that classifier.

In contrast, in stage 1 when we trained the encoder $E(\cdot)$ against the classifier $C_1(\cdot)$, we maximize the negative log-likelihood. The same problem does not occur here because it is $E(\cdot)$ and $G(\cdot, \cdot)$, not $C_1(\cdot)$, who tries to maximize it. Considering that the pixel values of the generator's output has a finite range, for a fixed classifier, the log-likelihood is bounded.

Indeed, in stage 1 the negative log-unlikelihood would not be a favourable choice: in the equilibrium, the classifier is expected to perform poorly (no better than random guess), at which point the negative log-unlikelihood would have tiny gradients while the negative log-likelihood would have large gradients.

**A Note on Using Classification Accuracy as Quality Measure.** From the discussion above, we conclude that the accuracy of classification by style on the generated samples with an independently trained classifier cannot properly assess the level of success of style transfer: since such a classifier, trained solely on real samples, cannot capture complete style information, a high accuracy achieved by generated samples does not indicate that the style transfer is successful. It is likely, though, that the generator is of bad quality if the accuracy is very low.

However, we feel that if the generator could score a high accuracy even against an adversarial classifier, it must be considered a success. Unfortunately, our method has yet to achieve this goal.

In table 8 we report the classification accuracy by both the adversarial and the non-adversarial classifier on samples generated both by the generator trained against the adversarial classifier and by the generator trained against the non-adversarial classifier. Given that for the majority of artists the number of artworks available is quite limited, to use the data at maximum efficiency we used all data for training and did not have a separate test set. As a substitute, we generate samples from content codes drawn randomly from the learned content distribution which is modeled by a multivariate normal distribution over the whole training set. Notice that during stage 2 training we only use $E(x)$ where $x$ is from the training set and $E(\cdot)$ is fixed, so the network only ever sees a fixed finite set of different content codes.

Table 8: Top-1 classification accuracy of two classifiers on samples generated from two generators

|  | Adversarial $C_2$ | Non-adversarial $C_2$ |
|---|---|---|
| $G$ trained with adversarial $C_2$ | 14.37% | 86.65% |
| $G$ trained with non-adversarial $C_2$ | 1.85% | 88.59% |

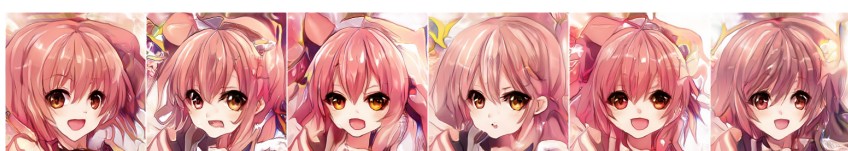

Figure 13: Images generated from fixed style and different contents, when explicit condition on content is removed.

Thus, random drawn content codes are indeed unseen which serves the same purpose as a separate test set.

In each test, the same number of samples as in the training set (106,814) are generated such that the content codes are independent and random and each artist appear exactly as many times as in the training set.

As can be seen, there is a huge discrepancy between the classification accuracy scored by the different classifiers. If trained to do so, the adversarial classifier can easily tell that the generated samples are not of the correct style, so extreme caution must be exercised if any conclusions is to be drawn from the classification accuracy by a non-adversarial classifier.

### C.3 EXPLICIT CONDITIONING ON CONTENT IN STAGE 2 GENERATOR

Our treatment of content in stage 2 is also quite different, in that we use a content loss to explicitly condition the image generation on content. Compare this to (Chou et al., 2018) where there is no explicit condition on content, rather, the stage 2 generator's output is added to the stage 1 decoder so that the stage 1 condition on content, which is guaranteed by minimizing the reconstruction loss, is inherited by stage 2.

While we did not experiment with their approach, we show here that in our framework the explicit condition on content by content loss is necessary. We repeat stage 2 training with $\lambda_{\text{cont}} = 0$ so that $E(G(E(x), S(a')))$ is not constrained to be close to $E(x)$. We then examine if style and content can still be controlled independently. Images generated by fixing the style and varying the content is shown in figure 13, while images generated by fixing the content and varying the style is shown in figure 14.

We can clearly see that the content code has lost most of its ability to actually control the content of the generated image, while the style code would now control both style and content. Curiously, the content code still seems to control the head pose of the generated character, as this is the most noticeable variation in figure 13 while in figure 14 the characters had essentially the same head pose.

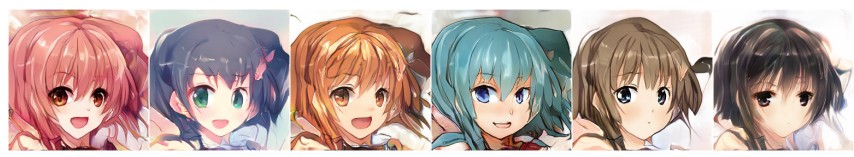

Figure 14: Images generated from fixed content and different styles, when explicit condition on content is removed.

This can be considered a form of "partial mode collapse": while the network is still able to generate samples with rich variation, some of the input variables have little effect on the generated image, and part of the input information is lost. This happens despite the fact that the stage 2 generator is initialized with the weights of the final stage 1 generator, in which the content code is able to control the content. So, this is not the case where the number of input variables is more than the dimension of the distribution of the dataset so that some input variables never got used. Rather, the ability to control content is initially present but subsequently lost. So, in our approach, an explicit condition on the content by a content loss is necessary.

