# OpenReview forum: "Disentangling Style and Content in Anime Illustrations"
_ICLR.cc/2020/Conference — Reject_

### Official Review · AnonReviewer2 · 2019-10-23
**Official Blind Review #2**

**Rating:** 3

**Review:**

The paper proposes an image generation method with the focus on generating anime faces from various artists. The proposed method which is a combination of conditional GANs and conditional VAEs manages to generate high fidelity anime images with various styles.

The paper is well-written and easy to follow and understand. The goals are clearly stated and the background (which is more of history), as well as related work, is comprehensive. The decision behind every design decision has been mentioned in detail which makes the paper stronger. The main writing flaw of the paper is in the figures where more annotation and caption is required to make them easy to understand. For example:
- The significance of colors in Figure 1 (architecture of the model) is not annotated at all and the letters are not clear either (although they are described in the text itself a figure should be comprehensive by itself).
- Figure 2 and Figure 4 are really hard to understand with very limited annotation and caption. I had to read the text multiple time to Figure out what is what in these figures which is not a good sign for clarity.

In terms of experiments, I think where the paper suffers the most is in comparison with other conditional methods. In Section 5 it has been clearly mentioned that "this result can be expected from a class-conditional GAN and the focus in on Disentanglement" however very little evidence has been provided for superior disentanglement. More experiments are required to demonstrate the capabilities of the model compared to other conditional methods (which is currently only limited to StarGAN) as well as its capability of disentanglement. I agree with the authors that quantitative evaluation of generated anime faces is not easy (although it is possible with a carefully designed human study), however, the disentanglement (which is the focus of the paper) is easy to evaluate quantitatively. This demands for more experiments on disentanglement datasets with known generative factors. Although the current ablation study in the Appendix provides more details for architectural decisions, a more qualitative and quantitative comprehensive ablation study (by actually ablating the final model) can help to demonstrate these decisions.

In conclusion, the paper has great results. We all know a big part of writing this kind of paper is to make the model "work" and authors truly demonstrate that they worked hard. However, the impact of the paper (in the current form) is not clear. With the focus on disentanglement, little evidence has been provided to justify the capability of the proposed method. I believe by addressing my comments on the experiments the paper can be easily pushed above the acceptance bar. Also releasing the code dataset should increase the impact of the paper.


**Experience Assessment:**

I have published one or two papers in this area.

**Review Assessment: Checking Correctness Of Derivations And Theory:**

I assessed the sensibility of the derivations and theory.

**Review Assessment: Checking Correctness Of Experiments:**

I carefully checked the experiments.

**Review Assessment: Thoroughness In Paper Reading:**

I read the paper at least twice and used my best judgement in assessing the paper.

---

> ### Author Response · Authors · 2019-11-14
> **Response to reviewer 2**
>
> We thank for the enthusiasm about our work and for pointing out at many strengths and appreciate the feedback.
>
> The problem of figures being unclear can be fixed within the discussion period.
>
> For the reviewer's other major concerns, it is not our own words that "the focus is on disentanglement", and the comment that "this result can be expected from a class-conditional GAN" is not made in regard to disentangling either. While disentanglement is a major goal, our conditional image generation part is also considerably different from existing methods and those differences are of equal importance to the success of the method.
>
> By "this result can be expected from a class-conditional GAN" we are referring to the fact that if a class-conditional GAN is trained with the label being the artist, then it provides the functionality of controlling the style of the generated image by altering the input artist label, which we also provide. What a typical class-conditional GAN cannot do is to guarantee that when the input artist label is altered the content remains unchanged. In our method, this is guaranteed by requiring that the generated image can be encoded back to the input content code. While this would benefit from a better disentangled encoder, it is a contribution in its own right.
>
> Our classifier in stage 2 is also different from a typical class-conditional GAN in that it is adversarial and is thus able to capture the most subtle aspects of an artist's style. The effectiveness of this improvement is also one place where we do prove qualitative evaluations.
>
> So the high-fidelity results we were able to obtain is due to the combined strength of many improvements. Should the reviewer demand more evaluations of the effectiveness of the disentangling step, we will run these additional experiments and update our results.
>
> We have prepared the code and training dataset for this work and we will release our code to the public upon acceptance of this work. We hope the score can be improved based on this rebuttal. We will improve the paper with these feedback and please let us know if there are any further questions.

---

> ### Author Response · Authors · 2019-11-15
> **Revision to the submission**
>
> We have added some quantitative evaluation in our updated submission. Please find the details in the official comments. Thanks.

---

### Official Review · AnonReviewer1 · 2019-10-25
**Official Blind Review #1**

**Rating:** 6

**Review:**

Contributions:
1. This paper proposes a method to learn disentangled style (artist) and content representations.
2. By carefully designing two-stage training objectives, the method learns a style-independent content-encoding E at the first stage and the style encoder S and generator G both from the first and the second stage.
3. Empirical results justify the validity of the method.

I think this paper makes a good contribution to disentangle style and content in anime. My main concern is the complicated learning procedure design may affect the reproducibility of this method. Moreover, I will suggest several points to the authors to clarify in the main text.

1. I encourage the authors to release their code when published.

2. In stage 1 (Style Independent Content-Encoding), the purpose of the classifier C, to my understanding, is to try to classify the generated example G(E(x), S(a')) as the "ground-truth" style (a). That is, the classifier C tries to disregard the S(a') when making a decision. As an adversarial player, E, G, S will try to fool C by making E(x) to be non-informative regarding the style. However, since you are still optimizing G and S, how do you make sure that it is safe to hold E fixed while still changing G and S in the second stage? Or more specifically, how do you make sure the style encoding network S preserves a good one in the second stage? Aside from that, are you using the trained G, S from the first stage to initialize G, S in the second stage?

3. There are eight different terms in stage 2, so it worth checking the necessity for those terms. E.g. what happens if you drop the L_cont term? The term L_cont seems to guarantee the validity of E, but E is fixed in step 2.

**Experience Assessment:**

I have read many papers in this area.

**Review Assessment: Checking Correctness Of Derivations And Theory:**

I carefully checked the derivations and theory.

**Review Assessment: Checking Correctness Of Experiments:**

I carefully checked the experiments.

**Review Assessment: Thoroughness In Paper Reading:**

I read the paper at least twice and used my best judgement in assessing the paper.

---

> ### Author Response · Authors · 2019-11-14
> **Response to reviewer 1**
>
> It is correct that in stage 1 the classifier $C$ tries to disregard $S(a')$ and tries to find information about the true author $a$ from $E(x)$, and $E$, $G$, and $S$ jointly tries to purge style information of $a$ from $E(x)$.
>
> It is true that $G$ and $S$ are changing in stage 2 while $E$ is fixed, but note that $G$ and $S$ don't change arbitrarily, they are still cooperating with $E$, so they should not make changes that cause the output of $E$ to be unsuitable. Furthermore, it is necessary to fix $E$: if we allow $E$ to change and at the same time minimize $||E(x)-E(G(E(x), S(a)))||$, then the most obvious way to do so is for $E$ to give degenerate output (e.g. encode everything to 0) which renders $E$ useless.
>
> In stage 2 $G$ and $S$ are initialized from the trained $G$ and $S$ in stage 1, but this is optional. It might take longer if $G$ and $S$ are trained from scratch in stage 2 but the end result should be comparable.
>
> Regarding the necessary of the many loss terms in stage 2: for a class-conditional GAN where the class condition is enforced by an auxiliary classifier in the discriminator, five terms is the bare minimum: discriminator's loss on real and generated samples, classifier's loss on real samples, and generator's loss against the discriminator and the classifier.
>
> Among the additional terms we added, $L_{cont}$ is necessary. It is not used to guarantee the validity of $E$. Instead, it is used ensure that the generator does actually use the content input in the intended manner, that is, it does generate images that has the content represented by the content code. The effect of dropping it is shown in section C.2 in the appendix.
>
> The classifier's loss on generated samples is not absolutely necessary, but it changes the behavior of the classifier towards the generated samples from passive to adversarial, which is a qualitative difference from prior works and is one of our contributions. The effect of dropping it is discussed in detail in section C.2.
>
> The last term, the KL-divergence loss on the output of $S$, is largely optional. It tries to constrain the distribution of style code and supposedly could benefit such things as interpolating between two styles. If this is not a concern, this term can be dropped.
>
> In short, seven of the eight terms are necessary.
>
> We have prepared the code and training dataset for this work and we are willing to release them if this work can be published.

---

### Official Review · AnonReviewer3 · 2019-10-29
**Official Blind Review #3**

**Rating:** 3

**Review:**

This work introduces a Generative Adversarial Disentangling Network based on two stages training the first aims at learning a style independent content encoder and then content and style conditional GANs is used for synthesis.
At stage 1 training to prevent the encoder from encoding authors introduce a gan style training in which an adversarial classifier that tries to predict the corresponding artist from the encoded image.
At stage 2 is training a style/content conditional gan. To condition on the style (artist) authors introduce an extra adversarial classifier so the generator tries to generate samples that would be classified as the artist that it is conditioned
on. While to condition on the input content another loss is ensuring that the generated image is encoded back to its content input.

Authors compare the proposed method against the original neural style transfer and StarGAN over various styles within the context of anime illustrations and the NIST Dataset where styles are being represented by artist name.

While the work tackles some of the problems by conditioning only on artist names other than style features that might be hard to have annotations for. The proposed modifications are quite incremental. Additionally, the experiments section is quite weak, evaluation is only done quantitatively over some cherry-picked examples, although some extra ablation study in the appendix is provided.



**Experience Assessment:**

I have read many papers in this area.

**Review Assessment: Checking Correctness Of Derivations And Theory:**

I carefully checked the derivations and theory.

**Review Assessment: Checking Correctness Of Experiments:**

I assessed the sensibility of the experiments.

**Review Assessment: Thoroughness In Paper Reading:**

I read the paper at least twice and used my best judgement in assessing the paper.

---

> ### Author Response · Authors · 2019-11-14
> **Response to reviewer 3**
>
> It may seem that our major contribution is tackling the scarcity of artworks reliably labelled by style, by using artist label instead. This is in fact a practical design choice based on the availability of training data and is of minor importance.
>
> The major contributions are:
>
> 1. Improving the disentangling of two factors of variation in stage 1 by observing that the instability of the output distribution of the encoder $E$ for the unlabelled factor (content) enables it to also encode information about the labelled factor (style) but avoid being correctly classified by the adversarial classifier $C$, and proposing a modification of data flow and training procedure to eliminate this instability, thus achieving a cleaner disentanglement.
>
> 2. In stage 2, on top of a class-conditional GAN, explicitly adding a loss term to condition on the input content code, so that the generator is guaranteed to generate images with the same content from the same content code in combination with different style codes.
>
> 3. In stage 2, adding a loss term to train the classifier to not classify generated images into the correct class, thus making the classifier adversarial against the generator, in contrast to previous works where the classifier is either cooperative or passive towards the generator. This has the effect of letting the classifier learn every aspect of the style of each artist, beyond what is enough to tell different artists apart.
>
> The impact of each of these changes is studied in a separate ablation study in appendix C.
>
> Besides these technical contributions, we made conceptual arguments on why statistics-based representations of style and domain-independent definitions of style does are problematic for general style transfer problems, thus providing a different viewpoint for these problems.
>
> For the experiments on NIST, we also demonstrated that our method works equally well when the labelled factor is the digit and the unlabelled factor is writer identity, giving an example where the labelled factor does not conceptually correspond to "style".
>
> Please let us know if there are any further concerns.

---

> ### Author Response · Authors · 2019-11-15
> **Revision to the submission**
>
> We have added some quantitative evaluation in our updated submission. Please find the details in the official comments. Thanks.

---

### Author Response · Authors · 2019-11-15
**Revision to the submission**

We have updated our submission, mainly to address reviewer 2 and 3's concern about the lack of quantitative evaluations. In particular, we augmented the experiments on the NIST dataset in appendix B with evaluations of the effectiveness of the disentangling encoder, in the new section B.4, exploiting the fact that in the NIST dataset both variations of interest are labelled. The numbers are computed from the exact same networks used for generating the visualizations and no additional training was done.

Besides this, we added reference to the new experiments in the main text, adjusted the caption of some figures for better clarity, and removed some text to keep the length within limits.

---

### Decision · Program_Chairs · 2019-12-19

**Decision:**

Reject

**Comment:**

This paper proposes a two-stage adversarial training approach for learning a disentangled representation of style and content of anime images. Unlike the previous style transfer work, here style is defined as the identity of a particular anime artist, rather than a set of uninterpretable style features. This allows the trained network to generate new anime images which have a particular content and are drawn in the style of a particular artist. While the approach works well, the reviewers voiced concerns about the method (overly complicated and somewhat incremental) and the quality of the experimental section (lack of good baselines and quantitative comparisons at least in terms of the disentanglement quality). It was also mentioned that releasing the code and the dataset would strengthen the appeal of the paper. While the authors have addressed some of the reviewers’ concerns, unfortunately it was not enough to persuade the reviewers to change their marks. Hence, I have to recommend a rejection.